# Imparting multi-functionality to covalent organic framework nanoparticles by the dual-ligand assistant encapsulation strategy

Liang Chen[1,2,4], Wenxing Wang[1,4], Jia Tian [3], Fanxing Bu[1], Tiancong Zhao[1], Minchao Liu[1], Runfeng Lin[1], Fan Zhang [1], Myongsoo Lee [1], Dongyuan Zhao [1] & Xiaomin Li [1✉]

The potential applications of covalent organic frameworks (COFs) can be further developed by encapsulating functional nanoparticles within the frameworks. However, the synthesis of monodispersed core@shell structured COF nanocomposites without agglomeration remains a significant challenge. Herein, we present a versatile dual-ligand assistant strategy for interfacial growth of COFs on the functional nanoparticles with abundant physicochemical properties. Regardless of the composition, geometry or surface properties of the core, the obtained core@shell structured nanocomposites with controllable shell-thickness are very uniform without agglomeration. The derived bowl-shape, yolk@shell, core@satellites@shell nanostructures can also be fabricated delicately. As a promising type of photosensitizer for photodynamic therapy (PDT), the porphyrin-based COFs were grown onto upconversion nanoparticles (UCNPs). With the assistance of the near-infrared (NIR) to visible optical property of UCNPs core and the intrinsic porosity of COF shell, the core@shell nano-composites can be applied as a nanoplatform for NIR-activated PDT with deep tissue penetration and chemotherapeutic drug delivery.

[1] Department of Chemistry, Laboratory of Advanced Materials and Shanghai Key Laboratory of Molecular Catalysis and Innovative Materials, State Key Laboratory of Molecular Engineering of Polymers, Fudan University, Shanghai, PR China. [2] Materdicine Lab, School of Life Sciences, Shanghai University, Shanghai 200444, PR China. [3] Department of Chemistry, University of Victoria, Victoria, BC, Canada. [4] These authors contributed equally: Liang Chen, Wenxing Wang. ✉email: lixm@fudan.edu.cn

A s an emerging class of porous crystalline materials, covalent organic frameworks (COFs) are featured with high and regular porosity, low crystal density, and excellent structural stability[1,2], which make them promising candidates for various applications in catalysis[3], gas storage and separation[4,5], sensors[6], and biomedicine[7–10]. The controllable fabrication of COFs in terms of their components, structure, and morphology is of utmost significance to achieve well-designed COFs with desirable properties for different applications. By serving as a unique host matrix for various functional species, COFs offer the opportunity to develop new types of hybrid materials with collective or enhanced properties in comparison to the pure COF counterparts, such as the unique catalytic, optical, and electrical properties[11].

The incorporation of inorganic functional entities in COFs is one of the most important strategies to construct the COFs nanocomposites, which attracts much attention due to the diverse physicochemical properties of those functional entities[12]. Generally, there are two routes to prepare the functional COF nanocomposites. The first one is the post-loading strategy in the pre-synthesized COFs. The functional entities are limited to ultrasmall metal or metal oxide nanoparticles[13–18]. The encapsulation of pre-synthesized functional nanoparticles during the formation of COFs is another commonly used strategy to obtain functional COF nanocomposites[19], i.e., the core@shell structure, in which functional cores can be encapsulated into COFs to endow it with abundant optical, electrical, or magnetic properties[20].

Although there are many reports on core@shell structured COF nanocomposites and their applications, this emerging area still confronts significant challenges. First, the variety of functional core in the core@shell structured COF nanocomposites is still limited[21,22]. Given that the surface properties of the functional cores are vastly different, the previously reported methods always involve the pre-grafting of COF monomers and subsequent harsh solvothermal reaction for the shell growth, which makes it difficult to control the nucleation and growth kinetics of COF shell on their surface[23,24]. Thus, in order to enrich the feature set and applications of COFs, it is highly desired to develop a general method for the synthesis of the core@shell structured COF nanocomposites. Second, the controllable synthesis of COF-based nanomaterials with uniform morphology, which is beneficial to finely regulating the structure-performance relationship[25], is an urgent yet quite challenging goal. In addition, the agglomeration of COF crystallites is a commonly observed complication due to the "unavoidable" crosslinked network among the COF nanoparticles[26]. Therefore, to maximally optimize the performance of the COF nanocomposites, it is also important to ensure that the nanocomposites are solution-processable and monodisperse without agglomeration[27].

Here, we report a general dual-ligand assistant strategy for the synthesis of the uniform inorganic nanoparticles functionalized COF nanocomposites. With this versatile strategy, the COFs can be endowed with abundant and unique optical, electrical, magnetic properties. The uniform core@shell, hollow, bowl-shape, yolk@shell, and core@satellites@shell nanostructures can be delicately fabricated and the shell thickness can be well tuned from ~10 to ~50 nm. Moreover, the composition of COF shell can also be varied by using different types of monomers. With the obtained inorganic nanoparticles functionalized COF nanocomposites, some predicaments for the applications of pure COFs can be overcome. For example, the porphyrin-based COFs have shown great potential as photosensitizers for photodynamic therapy (PDT), but it only can be activated by visible light with insufficient tissue penetration depth. This predicament is conquered by integrating upconversion nanoparticles (UCNPs) with the porphyrin-based COFs. With the merit of the near-infrared (NIR) to visible optical property of UCNPs core and the intrinsic porosity of COF shell, the synthesized core@shell nanocomposites is utilized as a multifunctional platform for synergistic NIR-activated PDT and chemotherapy, which can realize superior antitumor efficiency.

## Results

**Dual-ligand assistant strategy for core-shell structured COFs.** The uniform inorganic nanoparticles functionalized core@shell structured COF nanocomposites can be obtained via dual-ligand assistant strategy. The synthetic process can be divided into three parts (Fig. 1a): (1) the sequential adhesion of polyethyleneimine (PEI) and polyvinylpyrrolidone (PVP) ligands onto the surface of the initial nanoparticles, (2) the controllable assembly of the monomers of COF shell on the nanoparticles, and (3) the further growth and thermal crystallization of the amorphous shell.

As a model core, the dense SiO$_2$ nanospheres with a size of 120 nm were firstly modified with dual ligands of PEI and PVP sequentially (Supplementary Fig. 1). Two organic linkers, 1,3,5-tris(4-aminophenyl)benzene (TAPB) and 2,5-dimethoxyter-ephthalaldehyde (DMTP), were used as the monomers to form the COF shell. The uniform core@shell structured SiO$_2$@COF nanocomposites were synthesized by the proposed strategy. Transmission electron microcopy (TEM) image clearly shows that the COF shell are uniformly coated on the dual ligands co-modified SiO$_2$ nanospheres (Fig. 1b and Supplementary Fig. 2). The shell thickness is about ~39 nm, which can be well tuned from ~10 to ~50 nm (Supplementary Fig. 3). The high-resolution TEM (HRTEM) image reveals the highly crystalline nature with an interplanar spacing of ~2.8 nm, which corresponding to (100) planes of the TAPB-DMTP-COF (Fig. 1c, d)[28]. The selected area electron diffraction pattern shows a distinct ring and the calculated d-spacing matchs well with the result of HRTEM (Fig. 1e). Thus, it can be concluded that the 2D plane of the COF layers is parallel to the SiO$_2$ surface, and the fringes with an interplanar spacing of ~2.8 nm can be assigned to the periodic pores perpendicular to the 2D plane of the COF layers and the SiO$_2$ surface (Supplementary Fig. 4)[3,29]. Unlike the commonly observed agglomeration complication[30,31], the obtained core@-shell structured COF nanocomposites are very uniform without any aggregation, which can be demonstrated by the large area scanning electron microcopy (SEM) image (Fig. 1f) and the dynamic light scattering with a low polydispersity of 0.01 (Supplementary Fig. 5).

The Brunauer–Emmett–Teller surface area of the bare SiO$_2$ and core@shell structured SiO$_2$@COF are determined to be 39 and 435 m$^2$/g (Fig. 1g), respectively, which indicates the highly porosity of the COF shell. The periodic crystalline frameworks of the COF shell can be further demonstrated by X-ray diffraction (XRD) pattern (Fig. 1h). The strong diffraction peaks at 2.72, 4.81, 5.63, 7.39 and 9.65° can be clearly observed after the thermal treatment in the acetic acid condition, matching well with (100), (110), (200), (210), and (220) planes of TAPB-DMTP-COF, respectively[32]. It should be noticed that the shell is amorphous within the first 4 h reaction under room temperature, and the diffraction peaks emerge after only 2 h thermal treatment at 80 °C (Supplementary Fig. 6). The thermal crystallization process is crucial for the transformation of amorphous imine polymer shell into highly crystalline COF shell (Supplementary Figs. 7, 8). The small angle X-ray scattering (SAXS) also indicates the porous structure of prepared SiO$_2$@COF after thermal crystallization (Supplementary Fig. 9). In addition, the imine-linked COF was also characterized by Fourier transform infrared (FTIR) spectra

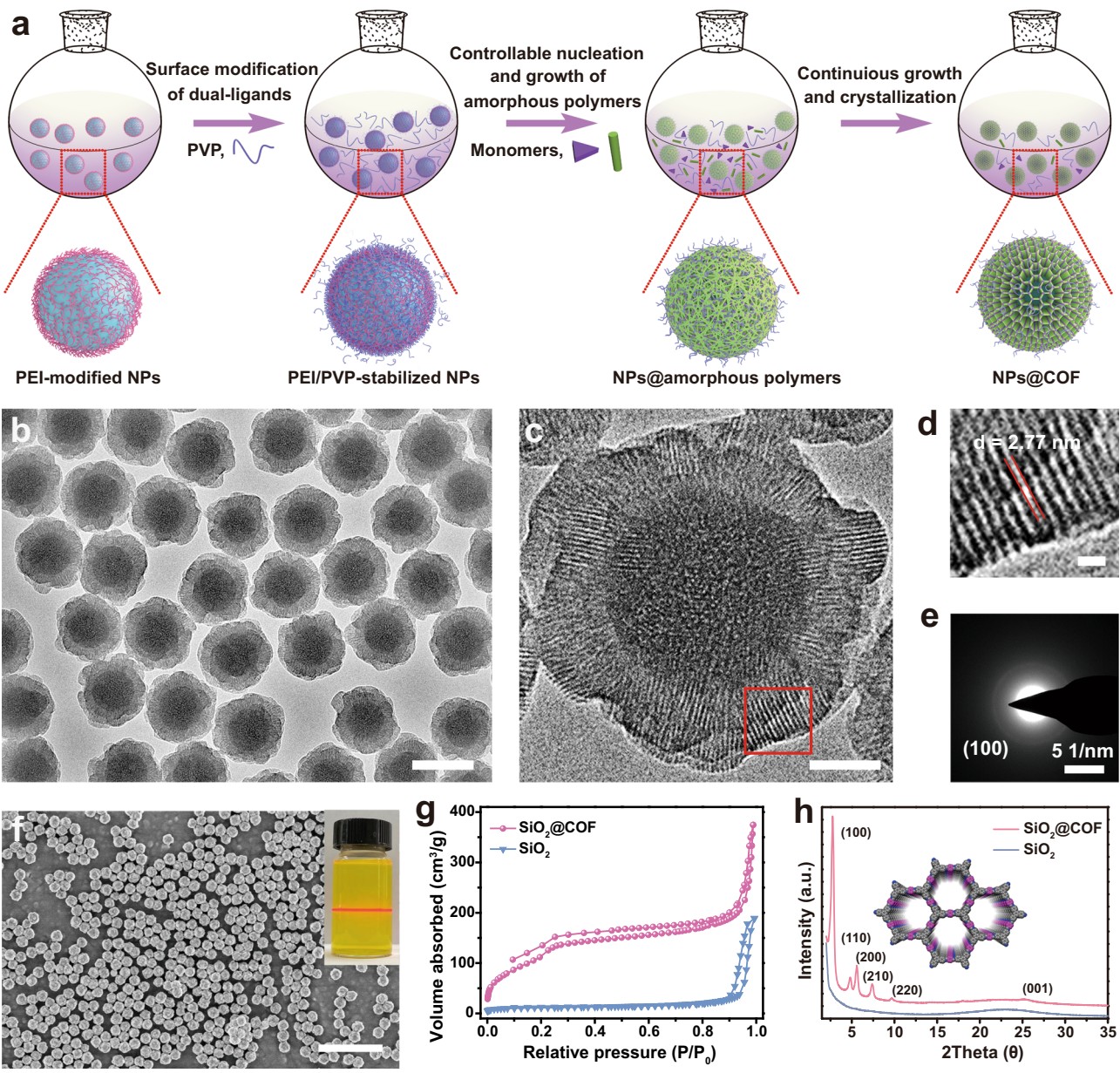

**Fig. 1 Synthesis and characterization of core-shell SiO₂@COF. a** Schematic illustration of the preparation of monodisperse COF-coated SiO₂ nanoparticles. **b**, **c** TEM images of the obtained SiO₂@COF nanocomposites. **d** HRTEM image of the COF shell from the area of red square in (**c**). **e** The corresponding selected area electron diffraction image taken from the COF shell. **f** SEM image of SiO₂@COF nanocomposites. Inset is the colloid dispersion of as-prepared SiO₂@COF after 1 month storage. **g** Nitrogen-sorption isotherms and (**h**) XRD patterns of the bare SiO₂ and SiO₂@COF nanocomposites. Scale bars are 200 nm in (**b**), 50 nm in (**c**), 5 nm in (**d**), and 1 μm in (**f**). A representative image of three replicates from each group is shown.

and $^{13}$C NMR spectrum. After coated COF onto SiO₂, the typical C=N stretching vibration at $1618\,cm^{-1}$ appears in the FTIR spectrum of SiO₂@COF, suggesting the formation of imine linkages in the framework of the COF shell (Supplementary Fig. 10). As shown in the solid-state $^{13}$C NMR, the peaks at 104, 111, 117, 123, 137, 150, and 203 ppm are assigned to the aromatic carbon signals from the COF shell (Supplementary Fig. 11). In particular, the peaks at around 150 ppm can be attributed to the carbon from the Schiff base bonds[33]. The peak at 171 ppm can be attributed to the carbonyl signal of PVP[34]. Apart from that, the optical property of COF-based materials has attracted increasing attention recently. For instance, Deng's group has reported a pioneer work of exploiting the photodynamic effect of pure COF by optimizing the molecular structure of the building blocks[35]. Thus, the band gap of SiO₂@COF was also estimated

here based on the UV–vis reflectance spectra to investigate the optical property of SiO₂@COF (Supplementary Fig. 12). Similar with the literature reports[36,37], the SiO₂@COF nanocomposites exhibit smaller band gaps compared with the corresponding molecular building blocks, demonstrating the cooperation between chromophores across the entire COF crystals.

**General applicability of the dual-ligand assistant strategy for functional integration.** Next, the broad applicability of this dual ligands assistant strategy is demonstrated by coating the COF shell onto functional cores with different morphologies and properties, which is crucial for enriching the feature sets and applications of COFs. As an example, the dendritic mesoporous silica nanoparticles (mSiO₂) can be uniformly encapsulated in COF shell to form the hierarchical porous structured

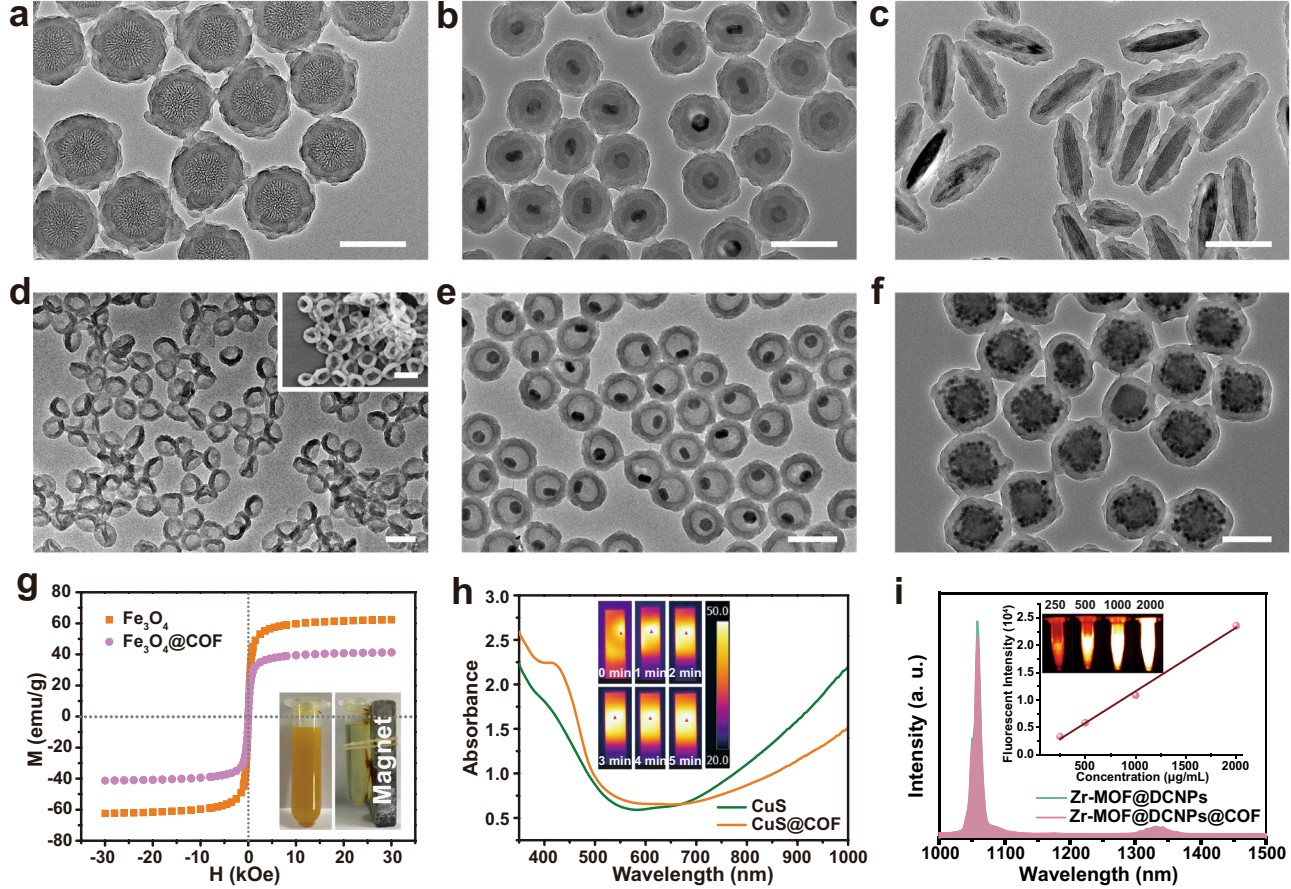

**Fig. 2 General applicability of the dual-ligand assistant strategy for synthesizing functional COF nanocomposites.** TEM images of COF-coated (**a**) mesoporous SiO$_2$ (mSiO$_2$), (**b**) Upconverting nanoparticles@SiO$_2$ (UCNPs@SiO$_2$), and (**c**) Fe$_2$O$_3$ ellipsoids. **d** TEM image of COF nanobowls obtained from SiO$_2$@COF after etching SiO$_2$ core. Inset is the SEM image of the obtained bowl-shaped COF. **e** TEM image of yolk-shell UCNP@COF obtained from UCNP@SiO$_2$@COF after etching the SiO$_2$ interlayer. **f** TEM image of core@satellite@shell structured Zr-MOF@DCNPs@COF. Scale bars are all 200 nm. **g** Hysteresis loops of the Fe$_3$O$_4$ and Fe$_3$O$_4$@COF nanocomposites. Inset represents the photos of Fe$_3$O$_4$@COF aqueous dispersion before and after separated by external magnetic field. **h** UV–vis spectra of CuS nanoplates and CuS@COF aqueous dispersions. Inset indicates the thermal images of CuS@COF dispersion after irradiated by 808 nm NIR laser (0.5 W/cm$^2$) for different time. **i** Downconversion luminescent spectra of Zr-MOF@DCNPs and Zr-MOF@DCNPs@COF dispersions. Inset represents the NIR II emission intensity of Zr-MOF@DCNPs@COF dispersions with different concentrations upon excited by 808 nm NIR laser and the corresponding NIR II images (concentration increases from left to right). A representative image of three replicates from each group is shown.

mSiO$_2$@COF nanocomposites (Fig. 2a). By using dense silica coated NaYF$_4$:Yb/Er@NaGdF$_4$ nanocrystals as the initial functional core, the monodisperse core@multi-shell structured NaYF$_4$:Yb/Er@NaGdF$_4$@SiO$_2$@COF nanocomposites with unique upconversion luminescence are also successfully fabricated (Fig. 2b). As silica can be readily coated onto various hydrophobic nanocrystals, it is expected that the dual ligands assistant strategy is also applicable for integrating other functional nanoparticles with COF. Furthermore, the silica can act as a sacrificial template to generate pure COF nanomaterials with unique nanostructures. For example, the SiO$_2$@COF can be transformed from a core@shell structure to the hollow structured COF nanoparticles after selectively etching the SiO$_2$ under the alkaline condition (Supplementary Fig. 13). Interestingly, when the thickness of COF shell is too thin to resist the capillary pressure[38], the hollow structure is collapsed to form bowl-shaped COF nanoparticles after the removal of SiO$_2$ core (Fig. 2d). In addition, yolk@shell structured UCNP@COF with a functional core in a large void space is also synthesized (Fig. 2e and Supplementary Fig. 14). The yolk-shell structured COF nanocomposites are expected to exhibit many advantages,

such as the high loading capacity, controllable releasing kinetics for cargos, unique spatial confinement effect for catalysis etc[39].

Apart from silica-based surface, the functional entities with other surface properties can also be integrated with COFs by this dual ligands assistant strategy. In previous works, complicated modification steps are required to introduce functional groups onto the surface of cores for the growth of COFs[40,41]. Here, the commonly used PEI and PVP ligands can be readily modified onto various nanomaterials to regulate the heterogeneous nucleation and growth of COF shell. As expected, the uniform COF shell can be coated onto Fe$_3$O$_4$ nanospheres, Fe$_2$O$_3$ ellipsoids, CuS nanoplates as well as gold nanoshells (Fig. 2c, Supplementary Fig. 15 and Fig. 16). The obtained COF-based nanocomposites also exhibit excellent magnetic or photothermal performance (Fig. 2g, h), suggesting the great potential of this strategy on imparting different functionalities to COF nanomaterials. In addition to the single functional component, the dual-functional core@satellites structured Zr-MOF@DCNPs (MOF = metal organic frameworks, DCNPs = downconversion nanoparticles NaGdF$_4$:Nd@NaGdF$_4$) is also successfully coated with

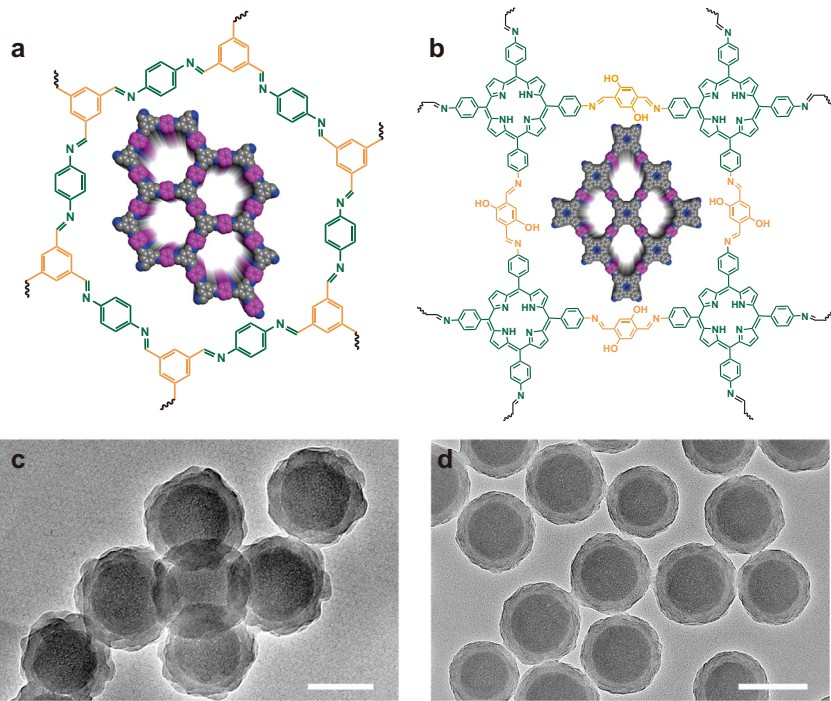

**Fig. 3 Dual-ligand assistant strategy for growing LZU-1 and porphyrin-based COF-coated SiO₂.** Structural unit and (inset) space filling model of (**a**) LZU-1 and (**b**) porphyrin-based COF. TEM images of the core@shell structured (**c**) SiO₂@LZU-1, and (**d**) SiO₂@porphyrin-COF nanocomposites. Scare bars are 100 nm in (**c**) and 200 nm in (**d**). A representative image of three replicates from each group is shown.

uniform COF shell (Fig. 2f). Elemental mapping of the Zr-MOF@DCNPs@COF nanocomposites shows that all the expected elements, including Zr from the MOF, Gd and Y from the DCNPs, O from ligand of MOF and C from MOF and COF, can be detected and match well with the core@satellites@shell nanostructure (Supplementary Fig. 17). Owing to the down-conversion luminescence of Nd-doped DCNPs, the NIR II emission of Zr-MOF@DCNPs@COF can also be detected under the excitation of 808 nm laser (Fig. 2j). Those results illustrate that the proposed dual ligands assistant strategy for the COFs encapsulation is applicable on various nanoparticles regardless of their composition, geometry, and surface property.

Next, 1,3,5-triformylbenzene (TFB) and 1,4-phenylenediamine (PDA), two typical monomers for synthesizing LZU-1[42], are selected to verify the applicability of this dual ligands assistant strategy for coating different types of COF shell. The result shows that the SiO₂ nanospheres are uniformly coated by LZU-1 (Fig. 3a, c). The XRD result of the obtained SiO₂@LZU-1 shows that the typical diffraction at 4.7° corresponding to (100) plane of LZU-1 can be detected (Supplementary Fig. 18), indicating the crystallinity of COF. By using the 5,10,15,20-tetrakis(4-aminophenyl)-21H,23H-porphine (TAPP) and 2,5-dihydroxylterephthalaldehyde as the monomers, the monodispersed core@shell nanocomposites with porphyrin-based COF shell are also successfully synthesized (Fig. 3b, d), and the crystallinity of porphyrin-based COF is also confirmed (Supplementary Fig. 19). It is worth mentioning that the XRD pattern of the SiO₂@porphyrin-COF is quite different from the COF-366 (even under the same synthesis conditions). It is speculated that crystalline of porphyrin COF on the spherical nanoparticles with a certain curvature could induce the weakened $\pi - \pi$ stacking between successive layers compared with bulk layered COFs, leading to their mismatch stacking. This mismatch stacking may further result in the changing of the cell parameters. This "curved surface induced abnormal crystallization" is very interesting, and we are still working on this part.

**Functional COF-based nanocomposites (UC-COF) for combined therapy.** The porphyrin-based COFs have shown great potential as nano-photosensitizers for PDT[43]. The use of such COFs in PDT, however, is limited by the poor tissue penetration depth of visible light. This predicament can be overcome by integrating UCNPs with the porphyrin-based COFs (Fig. 4a). As a proof of concept, a multifunctional nanoplatform with UCNPs@SiO₂ as core and porphyrin COF as shell is synthesized for tumor therapy. Because of the hydrophobic surface of the obtained UCNPs, a dense SiO₂ layer was coated on the UCNP to form the hydrophilic core@shell UCNP@SiO₂ nanoparticles, which greatly facilitates the further modification of dual ligands and the controlled growth of COF shell. The obtained UCNPs@SiO₂@COF nanocomposites (denoted as UC-COF) show uniform morphology with porphyrin-based COF shell (Fig. 4b). The UC-COF can be stabilized with PVP due to the interaction between PVP and imine units in the COF framework, endowing the synthesized UC-COF with good colloidal stability for biomedical applications (Supplementary Fig. 20). The upconversion luminescence of inner UCNPs functional core maintains very well after the encapsulation with the COF shell. Upon irradiated by 980-nm NIR laser, the UCNPs core can convert the NIR light to the visible light at ~ 545 and 655 nm (Supplementary Fig. 21), which can be absorbed by porphyrin-based COF shell and further induce the generation of reactive oxygen species (ROS) for PDT. Compared with the traditional visible light-triggered PDT, this UC-COF-based NIR-triggered PDT shows much deeper tissue penetration, which is more favorable for tumor treatment in deep tissue (Supplementary Fig. 22). The 1,3-diphenylisobenzofuran (DPBF) probe was used to detect the NIR-activated ROS generation. A significant reduction in the maximum absorption of DPBF can be observed under the irradiation of NIR laser (Supplementary Fig. 23), suggesting the high efficiency of NIR-activated PDT. The intracellular ROS generation in murine breast cancer cells (4T1 cells) was inspected by using 2,7-dichlorodihydrofluorescein diacetate (DCFH-DA) probe.

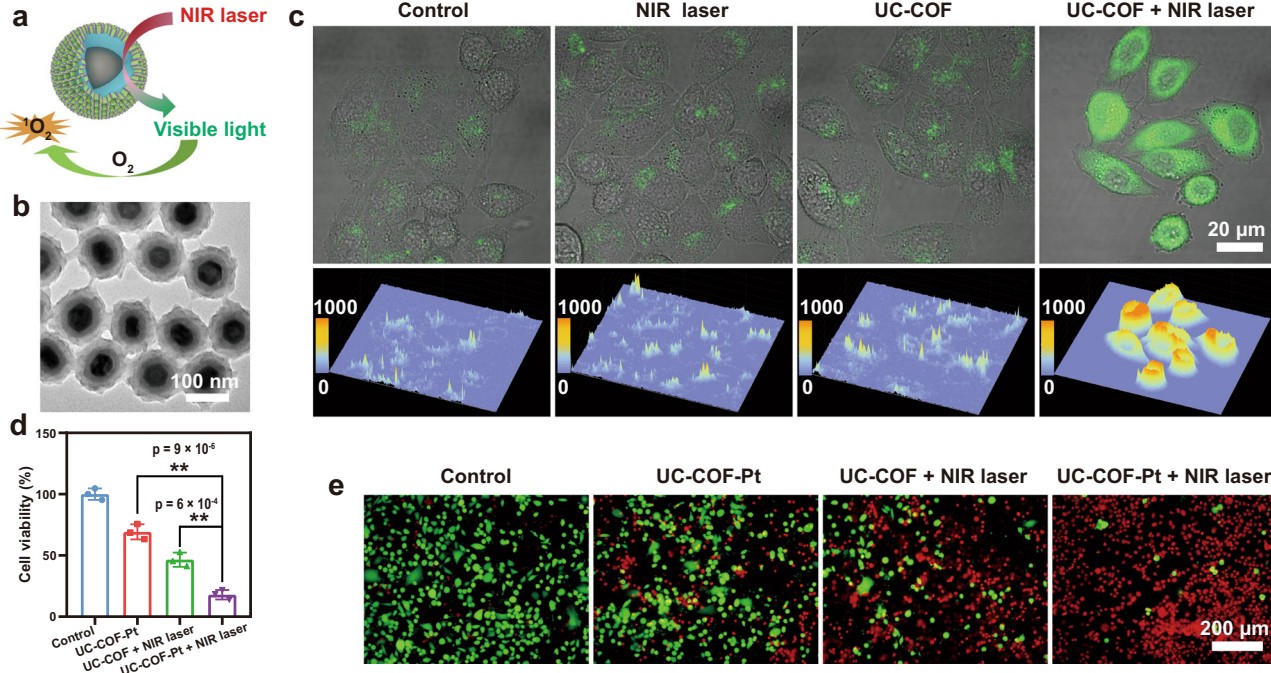

**Fig. 4 Synthesis of UC-COF for combined photodynamic and chemotherapeutic drug delivery. a** Schematic illustration of NIR laser-activated PDT by the porphyrin COF-coated UCNPs@SiO₂ (UC-COF). **b** TEM image of prepared UC-COF. **c** Confocal laser scanning microscope (CLSM) images of 4T1 cells and corresponding 3D surface plot images determined by the intensity of the green fluorescence after different treatments. A representative image of three replicates from each group is shown. **d** Cell viability of 4T1 cells and (**e**) the corresponding live-dead staining images by Calcein-AM and PI after different treatments, data in d are presented as mean ± s.d. derived from $n = 3$ independent biological samples. The statistical analysis was performed using one-way analysis of variance (ANOVA), followed by post hoc Tukey's method. Asterisks indicate a significant difference (*$p < 0.05$ and **$p < 0.01$). Source data underlying (**d**) are provided as a Source Data file.

Similar with the control group, the green fluorescence from the oxidized DCFH is very weak in cells treated with NIR laser or UC-COF alone (Fig. 3c). In sharp contrast, bright green fluorescence can be observed in the cells exposed to UC-COF under the NIR irradiation (Fig. 4c and Supplementary Fig. 24), which indicates that the intracellular ROS level is rapidly elevated due to the NIR-activated ROS generation.

Benefiting from the specific transmembrane mechanism of nanomaterials, the UC-COF can be easily internalized by 4T1 cells. Confocal laser scanning microscope (CLSM) images exhibit that the cell nuclei are surrounded by the red fluorescence from porphyrin of UC-COF (Supplementary Fig. 25), indicating the highly efficient endocytosis of the nanocomposites. The bare UC-COF exhibits negligible cytotoxicity against 4T1 cells and human umbilical vein endothelial cells (HUVECs), even at a high concentration of 200 μg/mL (Supplementary Fig. 26). The result of hemolytic assay also demonstrates that the negligible hemolytic effect of synthesized UC-COF (Supplementary Fig. 27), suggesting the excellent biocompatibility of UC-COF.

Further combined with the intrinsic pores of COF shell, the cisplatin is selected as a model drug for the synergistic chemotherapy with the NIR-activated PDT. The element mapping of cisplatin-loaded UC-COF (UC-COF-Pt) shows the evenly distributed Pt element in the COF shell (Supplementary Fig. 28). The loading capacity of cisplatin in this unique UC-COF nanocarriers is about 47 μg/mg according to the result of inductively coupled plasma atomic emission spectrometry (ICP-AES). 4T1 cells were subjected to different treatments to evaluate the cancer cell killing efficiency. After the treatments, the group of UC-COF-Pt under NIR irradiation shows the most efficient cell killing effect with a mortality of

~82.1% (Fig. 4d), which is much higher than the group of UC-COF-Pt (~ 31.8%) without the NIR light treatment and the mere PDT therapy of UC-COF (~53.5%). The live-dead staining images also demonstrate that more living cells exist in the single chemotherapy or PDT groups than that in the group of combined therapy (Fig. 4e). Thus, the prepared UC-COF nanocarriers can significantly improve the therapeutic efficiency by rationally integrating the NIR-triggered PDT and intracellular drug delivery.

Last, 4T1 tumor was subcutaneously xenografted on female Balb/c mice to assess the in vivo therapeutic efficiency. The tumor-bearing mice were randomly divided into five groups for different treatments (Fig. 5a): Control, NIR laser alone, UC-COF-Pt, UC-COF + NIR laser, and UC-COF-Pt + NIR laser. Remarkably, rapid tumor growth is observed in the former two group, while the tumor growth of mice treated with UC-COF-Pt is moderately inhibited (Fig. 5b). With the presence of NIR laser and chemotherapeutic drugs, the synergistic therapy exhibits the strongest inhibition effect compared with other groups (Fig. 5c, d). The tendency is also confirmed by the hematoxylin & eosin (H&E) and Tunel staining analysis of tumor slices (Fig. 5e). Histological analysis further reveals that vast areas of cavities and lots of overflowed cytoplasm can be observed in the tumor tissues from the UC-COF-Pt + laser group, which clearly implies the apoptotic or necrotic of tumor cells. Consistent with the above tendency, UC-COF-Pt + laser exhibits the highest antitumor efficiency due to the synergy of the NIR-triggered PDT and chemotherapy. The body weight of treated mice shows normal fluctuations, and no remarkable tissue damage or any other side effect are observed on heart, liver, spleen, lung, kidney (Supplementary Figs. 29, 30), confirming the good biocompatibility of UC-COF and well-tolerance of implemented treatments.

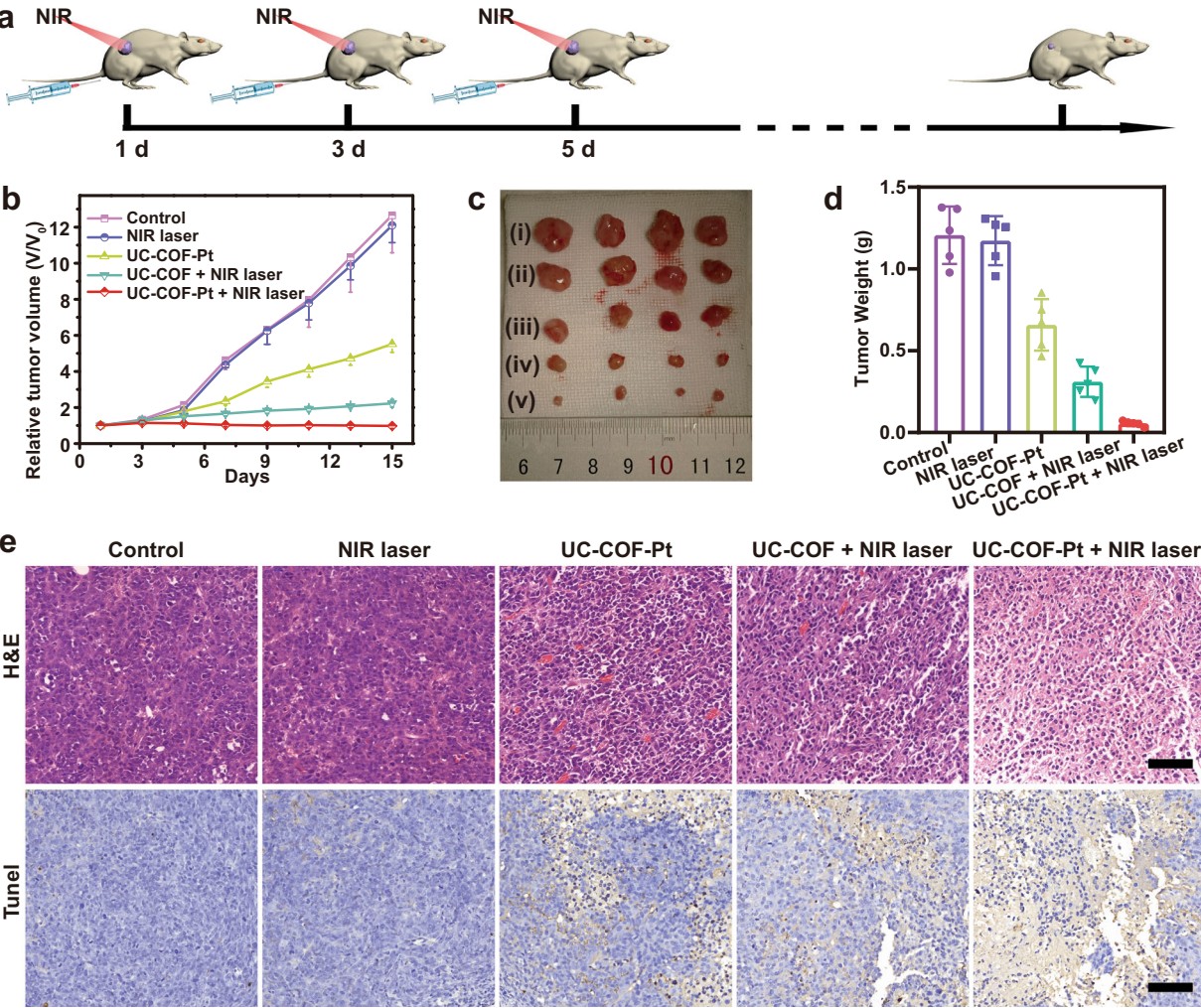

**Fig. 5 In vivo antitumor efficiency of UC-COF on 4T1-tumor-bearing mice. a** Schematic illustration of the building of subcutaneous tumor model and NIR light-triggered combinational therapy. **b** Tumor volume change of experimented mice under different treatments. **c** Photographs (i: Control, ii: NIR laser, iii: UC-COF-Pt, iv: UC-COF + laser, and v: UC-COF-Pt + laser), (**d**) weight, and (**e**) immunohistochemical analysis of extracted tumors from different groups after treatments. Scale bars in (**e**) are 200 μm. Data in (**b**) and (**d**) are presented as mean ± s.d. derived from $n = 5$ independent biological mice. Source data underlying (**b**) and (**d**) are provided as a Source Data file.

## Discussion

In our synthesis, the as-prepared cores were first modified with PEI and dispersed in the PVP solution, during which the PVP can attach onto the surface of cores (Supplementary Fig. 31). Thereafter, upon the addition of monomers of the COF shell, the polymerization of the monomers initiates to generate oligomers in solution under the catalysis of acetic acid. The homogeneous nucleation for the formation of the pure COF nanoparticles and heterogeneous nucleation on the surface of functional cores for the formation of core@shell structure have a competitive relationship, which is closely related to the corresponding critical nucleation concentration ($C_x$) (Fig. 6a). Generally, the critical nucleation concentration for the homogeneous nucleation ($C_5$) is constant under the predetermined conditions[44]. In comparison, the critical nucleation concentration of the heterogeneous nucleation is greatly influenced by the surface properties of the cores[45], which can be adjusted by changing the surface ligands. For instance, the weak interaction between COF oligomers and SiO₂ nanospheres with naked surface means the high critical concentration of heterogeneous nucleation ($C_4$) on the bare SiO₂ surface (even close to or higher than $C_5$), which led to the hard heterogeneous nucleation on the bare SiO₂ surface, and the generation of phase-separated pure COF nanoparticles (Supplementary Fig. 32).

When PEI is modified onto SiO₂ surface, the aldehyde monomers can be readily enriched onto the core surface through Schiff base reaction (Fig. 6b), which dramatically lowers the critical concentration of heterogeneous nucleation ($C_1$). In this situation, the oligomers rapidly deposit and grow on the surface of cores to form core@shell structure. However, too fast nucleation and growth rate inevitably give rise to the agglomeration of particles during the coating process. Similarly, PVP can also enrich the monomers onto the core surface on account of the hydrogen bonds between the PVP and aniline monomers (Fig. 6b), but this effect is much weaker than PEI. In the case of PVP modified SiO₂, the thickness of the COF shell is obviously thinner than that on the PEI-modified SiO₂ (Supplementary Fig. 33). Although the COF shells can be coated onto SiO₂ nanospheres, but this coating is the nonuniform coating. It can be observed that the COF domains on the SiO₂ nanospheres are "islands", which result in the irregular morphology of COF shell and even partially naked surface of the core. On the other hand, it is very difficult to tune the shell thickness by adjusting the PVP concentration.

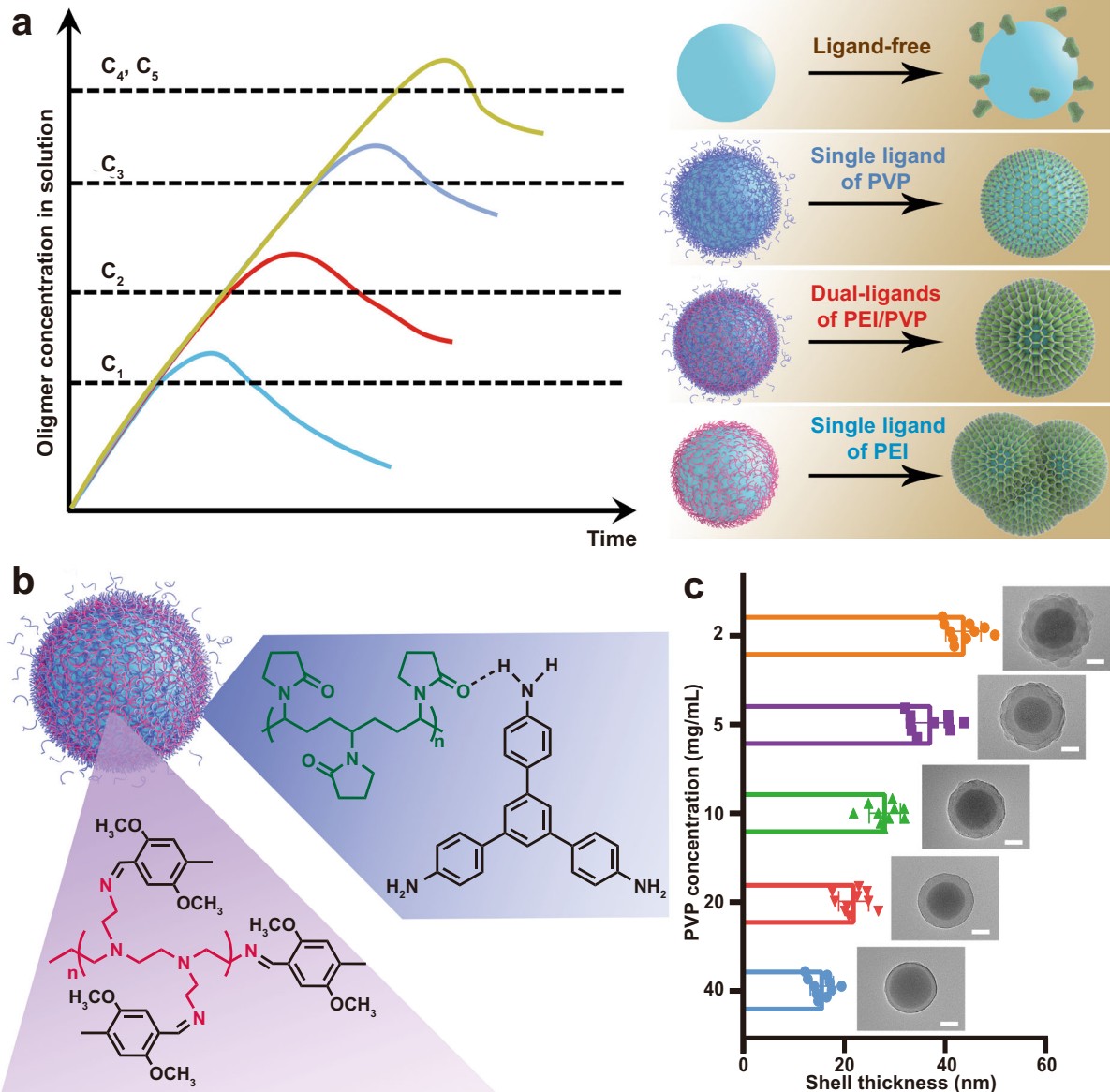

**Fig. 6 The mechanism of the dual-ligand assistant strategy. a** The schematic illustration of the oligomers concentration in the solution for the formation of the COF shell varied with the reaction time in the presence of the nanoparticles with different ligands (PVP, PEI and PVP, and PEI) based on LaMer model. $C_1$, $C_2$, and $C_3$ are the critical heterogeneous nucleation concentrations of polyimine oligomers on PEI-modified $SiO_2$ nanoparticles, PEI and PVP co-modified $SiO_2$ nanoparticles, and PVP modified $SiO_2$ nanoparticles, respectively. $C_4$ and $C_5$ represent the critical homogeneous and heterogeneous nucleation concentrations of polyimine oligomers on bare $SiO_2$ nanoparticles. **b** The schematic illustration of the interaction between the dual ligands and monomers. **c** The relationship between PVP concentration and shell thickness of the core@shell $SiO_2$@COF nanoparticles by dual ligands assistant strategy. Scale bars are 50 nm. Data in (**c**) are presented as mean ± s.d. derived from $n = 10$ independent particles. Source data underlying (**c**) are provided as a Source Data file.

Based on the above principles, the dual-ligand assistant strategy is proposed for coating uniform COF shell onto functional cores. By combining the dual ligands of PEI and PVP on the surface, the critical heterogeneous nucleation concentration of COF oligomers on the surface of cores ($C_2$) can be finely regulated, thereby avoiding the "too fast" or "too slow" nucleation and growth on the PEI or PVP single ligand modified surface. The kinetics of surface nucleation and growth is strongly associated with the ratio of the dual ligands. An appropriate amount of PVP can optimize the critical heterogeneous nucleation concentration to allow the moderate heterogeneous growth of COF shell on the cores without any phase-separation and agglomeration. When PVP are co-modified on the surface, the over-speed growth on the PEI single ligand modified surface can be greatly suppressed. The

decelerating growth speed of the COF shell can be attributed to strong interaction between the PVP and the protonated imine units (under acidic condition) in the COF framework, which further results in the passivation of the COF shell[46]. The growth speed of the COF shell can be well manipulated by tuning the ratio between the PEI and PVP. The shell thickness also can be adjusted from ~16 to ~45 nm by tuning the ratio between the dual ligands (Fig. 6c and Supplementary Fig. 34).

In summary, we have demonstrated a versatile approach for the growth of COF onto various nanoparticles, including $SiO_2$, metal oxides, semiconductor sulfides, and MOFs. The nucleation and growth kinetics of COF shell can be well manipulated by the dual ligands (PEI and PVP) on the surface of the functional nanoparticles. The proposed dual-ligand assistant strategy for the

construction of core@shell structured COF nanocomposites shows very broad applicability in regardless of the morphology, composition, or geometry of the cores as well as the types of monomers. The diverse morphologies of COF-based nanomaterials, including bowl-shape, hollow, and core@satellites@shell structure, have been successfully fabricated. More importantly, many specific properties (magnetic or optical) can be readily imparted into COF by the dual-ligand assisted strategy. Taking the advantages of NIR-to-visible upconversion luminescence and highly porous structure of COF, the prepared UC-COF can provide a unique nanoplatform for NIR-activated PDT and chemotherapy, realizing favorable antitumor efficiency. Overall, it is expected that this universal strategy can facilitate the construction of multicomponent COF-based nanostructures with collective functionalities for various applications.

## Methods

**Synthesis of monodispersed core@shell structured SiO$_2$@COF**. In a typical process, 10.0 mg of PEI-modified SiO$_2$ was added into 20.0 mL of anhydrous acetonitrile containing 40.0 mg of PVP. The dispersion was treated with ultrasonic for more than 0.5 h. Then, 4.0 mg of DMTP and 4.8 mg of TAPB were added. After 5 min stirring, 200 μL of glacial acetic acid was further added. The reaction was maintained at room temperature for 4 h. Afterward, 800 μL of acetic acid was further added and the mixture was heated to 80 °C for another 12 h reaction. The products were obtained and washed by centrifugation. The amount of PVP (100, 200, 400, and 800 mg) and monomers (0.5, 1, 2, 4, 8, and 16 mg of DMTP) were adjusted to obtain SiO$_2$@COF with different shell thickness. Other functional cores were also coated with COF shell according to the same procedures by replacing the corresponding seeds.

**Synthesis of UC-COF**. Briefly, 15.0 mg of the PEI-modified NaYF$_4$:Yb/Er@NaGdF$_4$@SiO$_2$ nanoparticles were added into 10.0 mL of o-dichlorobenzene/n-butyl alcohol (v/v = 1:1) with 100 mg of PVP dissolved, the dispersion was treated with ultrasonic for more than 0.5 h. Then, 7.5 mg of TAPP and 4.1 mg of 2,5-dihydroxy-1,4-benzenedicarboxaldehyde were added. After 5 min stirring, 0.10 mL of glacial acetic acid was further added. The reaction is maintained at room temperature for 4 h. Afterward, 0.30 mL of acetic acid was further added and the mixture was heated to 120 °C under nitrogen atmosphere for another 24 h reaction. Last, the products were obtained and washed by centrifugation.

**Intracellular ROS detection**. 4T1 cells and HUVECs were commercially provided by the cell bank of Chinese academy of science (Shanghai, China). 4T1 cells were seeded in a glass-bottom dish and incubated with UC-COF (50 and 100 μg/mL) for 3 h. Then the cells were washed with PBS and further cultured in fresh medium containing DCFH-DA for 30 min. Afterward, the cells were exposed to 980 nm laser for 15 min and washed for the CLSM observation. Moreover, the quantitative analysis of the fluorescence signal from the cells can also be determined by flow cytometry.

**Evaluation of in vitro therapeutic efficacy**. 4T1 cells were seeded on a 96-well plate at a density of 10$^4$ cells/well. After that, the cells were incubated with free UC-COF and UC-COF-Pt for 4 h. Subsequently, the materials were completely removed and the cells were washed with PBS. The groups of UC-COF and UC-COF-Pt were exposed to 980 nm laser (1 W/cm$^2$) for 15 min. Then, cells were rinsed with PBS for several times and incubated for another 20 h. Finally, the cell viability was evaluated by the standard CCK-8 assay. Moreover, the cells were also stained with Calcein-AM/PI after different treatments and directly observed by CLSM.

**In vivo antitumor effect**. All the animal experiments were approved by the Shanghai Science and Technology Committee (SYXK2014-0029, Shanghai, China) and performed in agreement with the guidelines of the Department of Laboratory Animal Science, Fudan University. 4–6-week-old Female Balb/c mice were brought from Slac Laboratory Animal Co. Ltd. (Shanghai, China). 4T1 cells suspended in FBS (4 × 10$^6$ cells) were subcutaneously injected into the right back leg of mice. When the size of tumors reached ~5 mm in diameter, the tumor-bearing mice were randomly divided into five groups (n = 5 for each group): control group, NIR laser, UC-COF-Pt, UC-COF + NIR laser, and UC-COF-Pt + NIR laser. For the treatments, the corresponding samples were intravenously injected, and 980 nm laser irradiation (1 W/cm$^2$, 15 min) was performed at 24 h post-injection (3 min irradiation with an interval of 1 min). The treatments were conducted at days 1, 3, and 5. The tumor size and body weight of the experimental mice were monitored every 2 days in subsequent 2 weeks. Additionally, the tumor tissues of each group were excised after the treatments for H&E and Tunel staining.

**Reporting summary**. Further information on research design is available in the Nature Research Reporting Summary linked to this article.

## Data availability

All relevant data supporting the findings of this study are included within the article and Supplementary Information files. Any other data are available from the authors upon reasonable request. Source data are provided with this paper.

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

## Acknowledgements

The work was supported by the National Key R&D Program of China (2018YFE0201701 and 2018YFA0209401), National Natural Science Foundation of China (22075049, 21875043, 21701027, 21733003, 21905052, 51961145403, and 22088101), Key Basic Research Program of Science and Technology Commission of Shanghai Municipality (17JC1400100), Natural Science Foundation of Shanghai (18ZR1404600, 20490710600), Shanghai Rising-Star Program (20QA1401200) and China Postdoctoral Science Foundation (Grant Nos. 2018M641911, 2018M630397).

## Author contributions

X.L. and L.C. contributed to the conception and design of the experiments, analysis of the data, and writing the paper. L.C. conducted the synthesis and characterization of the samples. W.W. and L.C. conducted the cell and animal experiment. F.Z., M.L., D.Z., and X.L. provided scientific guidance on the study design and experiments. J.T., F.B., T.Z., M.L., R.L., assisted L.C. for the synthesis of materials and the data collection and analysis. All authors contributed to the discussion and paper preparation.

## Competing interests

The authors declare no competing interests.
