## [Peer Review File · Nature Communications]

REVIEWER COMMENTS

Reviewer #1 (Remarks to the Author):

Li et al have reported a versatile approach to fabricate COFs onto various nanoparticles. The proposed dual-ligands (PEI and PVP) assistant strategy can be applied for the construction of COF nanocomposites with various structures (core@shell, hollow, bowl-shape, yolk@shell, etc.) regardless of the morphology, composition, or geometry of the cores as well as the types of monomers. Furthermore, many specific properties can be imparted into COF by embedding corresponding inorganic cores. For example, the authors synthesized a unique nanoplatform for NIR-activated PDT and chemotherapy by fabricating porphyrin-based COF on the upconversion nanoparticles (UCNPs). Finally, a reasonable mechanism of the method has also been proposed. However, some problems are needed to be addressed before considering the publication.

1) The authors claim that this method can be applied for coating different types of COF shell. However, it seems that the LZU-1 coated on the SiO₂ nanospheres is amorphous as shown in Figure S13, which should not be called "COF" anymore. Besides, the PXRD pattern of COF-366 synthesized via this approach (Figure S14) is significantly different from the previously reported result (Chem. Mater. 2011, 23, 18, 4094–4097). Therefore, it is dubious that this method is general for various COFs, or just applicable for limited kinds of COFs such as COF-OMe, one of the most highly crystalline COF.

2) More characterizations such as FT-IR and solid-state ¹³C NMR should be applied for providing more information on chemical structures of COFs, especially the imine linkage.

3) In the tumor therapy part, comparative experiments using porphyrin-based COF without UCNPs should be performed to tell the necessity of the UCNPs core and the importance of the method.

4) Some notes (e.g. Figure S5, S6, S8, and 3c) are not corresponding to the right figures, which should be double-checked and revised.

5) There are some grammar mistakes. Page 5, line 4, "be overcame" should be revised as "be overcome"; page 7, line 8, "reveal" should be revised as "reveals".

Reviewer #2 (Remarks to the Author):

This manuscript proposed a versatile dual-ligands assistant encapsulation strategy for the synthesis of COF-based nanocomposites with customizable structures and functional integration, which provides advancement in the preparation of inorganic nanomaterials functionalized COF nanocomposites. By precisely manipulating the nucleation and growth kinetics of COF, diverse structures of COF-based nanocomposites, including hollow, bowl-shape, yolk-shell, and core-satellites-shell, were successfully fabricated. The COFs were endowed with abundant optical, electrical and magnetic properties. By developing the UC-COF with unique NIR to visible upconverting luminescence, the predicament faced by tradition porphyrin-COFs in the visible light activated PDT with insufficient tissue penetration depth was overcome. Overall, this work has been well-organized, and the results are very interesting, which may attract broad interests from

multidisciplinary research communities for different applications. Therefore, it could be considered for publication after addressing the following issues.

1. The dual-ligands clearly played pivotal role in regulating the critical nucleation concentration of COF shell on the surface of SiO₂. Did the dual-ligands also affect the growth rate? What is the exact mechanism of controlling shell thickness by the ligand concentration as shown in Figure 6c?
2. The critical nucleation concentration was regulated by tuning the interaction between monomers and SiO₂ surface. Whether the concentration of single ligand (PEI or PVP alone) could also regulate the critical nucleation concentration?
3. Could the modification sequence of the dual ligands affect the final products? Please provide detailed description in the text.
4. The bowl and yolk-shell structure were obtained by selecting etching of SiO₂ segment. The author should investigate whether the porous structure of COF shell was changed or not after the alkaline etching process.
5. The UC-NPs were also pre-coated with SiO₂ for the synthesis of UC-COF, but SiO₂ seemed to be useless for the biomedical application of UC-COF. The authors should give a brief explanation about this.
6. The colloidal stability of UC-COF in different media should be evaluated by DLS.
7. The authors need to check the full text carefully, e.g., the caption for Figure 2f is missing, in the caption of Figure 5, "Scale bars in d" should be "Scale bars in e".

Reviewer #3 (Remarks to the Author):

This is another excellent work from the mesopore group. The authors target the key challenges in the composition of COFs with inorganic functional materials, i.e. the mismatch at the interface of these two different classes of materials. Different from previous works that using surface modification of small organic molecules, the method used here involves the combinational use of two types of polymers, PEI rich in amino functional groups, and PVP rich in aldehyde. These two polymers offer strong chemical interactions with two building blocks of the COF separately, therefore generate a robust interface for the adhesion and growth of COF coating layer. The advantage of using two kinds of polymers, instead of small molecules with the same functional group, is that the conformation of the polymers avoids intermolecular reaction between the amino and aldehyde functional groups, thus providing sufficient anchors for the COF building blocks at the interface. This combined with the relatively well established polymer coating on inorganic functional materials, such as metal oxides, makes this new method applicable for the composition of COF with a large variety of inorganic cores. In general, this is an important advance in the synthesis of state-of-the-art COF based functional materials, and it is highly recommended for publication. The following suggestions might help to further the quality of this work.

1. Is there any way to quantify the thickness of the polymer layers applied prior to the growth of COFs? This is critical for the reproduction of results and the quality control.
2. The orientation of the COF crystals should be further discussed. TEM figures only provide the projected view of each nanoparticle. The fringes in the high resolution images assigned to the COF crystal, could either reflect the interlayer distance between the layers of this 2D COF, or the size of the vertical pores perpendicular to the 2D plane of the COF layers, depending on the orientation of the COFs. Both of these two possibilities should be assessed and further studied. The orientation of the pores, perpendicular, or parallel to the spherical particle surface will influence its uptake behavior for the guest molecules.

3. It is known that the domain size of COF crystals and band structure are important for the optical behavior (Angew. Chem. Int. Ed., 2019, 58, 14213, Matter, 2020, 2, 4, 1049, Aggregate, 2021, <https://doi.org/10.1002/agt2.24>). Investigation on these molecular aspects of the COF coating will help to further understand the optical properties of the COF composite.

4. Another interesting aspect is the capability to produce Yolk shell shaped COF composite. It might worth of discussing the advantage of such structures with large inside cavity and small pore apertures in the shell.

Manuscript Title: Imparting multi-functionality to covalent organic framework nanoparticles by the dual-ligands assistant encapsulation strategy

Manuscript ID: NCOMMS-21-03389-T

Point-by-Point Response to Referees

Reviewer #1:

Li et al have reported a versatile approach to fabricate COFs onto various nanoparticles. The proposed dual-ligands (PEI and PVP) assistant strategy can be applied for the construction of COF nanocomposites with various structures (core@shell, hollow, bowl-shape, yolk@shell, etc.) regardless of the morphology, composition, or geometry of the cores as well as the types of monomers. Furthermore, many specific properties can be imparted into COF by embedding corresponding inorganic cores. For example, the authors synthesized a unique nanopatform for NIR-activated PDT and chemotherapy by fabricating porphyrin-based COF on the upconversion nanoparticles (UCNPs). Finally, a reasonable mechanism of the method has also been proposed. However, some problems are needed to be addressed before considering the publication.

Response: We appreciate the reviewer very much for the positive comments.

1. The authors claim that this method can be applied for coating different types of COF shell. However, it seems that the LZU-1 coated on the SiO₂ nanospheres is amorphous as shown in Figure S13, which should not be called “COF” anymore. Besides, the PXRD pattern of COF-366 synthesized via this approach (Figure S14) is significantly different from the previously reported result (Chem. Mater. 2011, 23, 18, 4094–4097). Therefore, it is dubious that this method is general for various COFs, or just applicable for limited kinds of COFs such as COF-OMe, one of the most highly crystalline COF.

Response: We thank the reviewer very much for this insightful comment. In our previous version of the manuscript, the LZU-1 coated on the SiO₂ nanospheres is amorphous. As we revise the manuscript, we try to explore ways to crystallize the COF shell layer through post-crystallization strategy. According to the literature reports, the crystalline degree of COF is greatly related to the solvents for the crystallization (*J. Am. Chem. Soc.* 2019, 141, 18271–18277). We found that the crystalline LZU-1 COF shell can be formed on the SiO₂ nanoparticles after prolonged thermal crystallization time in 1,2-dichlorobenzene/n-butanol. The results show that the uniform core@shell structured SiO₂@LZU-1 also can be obtained in the co-solvent of 1,2-dichlorobenzene/n-butanol after the post thermal crystallization for 5 days). The XRD pattern of the obtained SiO₂@LZU-1 shows the typical diffraction peak at 4.7° (**Figure S18**), corresponding to (100) plane of LZU-1. The related results and synthetic procedure for the SiO₂@LZU-1 have been provided in the revised manuscript.

With regard to porphyrin-based COF, the PXRD pattern is indeed different from the COF-366 although the strong diffraction peaks can be clearly observed. Actually, the underlying mechanism still remains unknown. It is speculated that crystalline of porphyrin COF on the spherical nanoparticles with a certain curvature could induce the weakened π - π stacking between successive layers compared with bulk layered COFs, leading to their mismatch

stacking. This mismatch stacking may further result in the changing of the cell parameters. This “curved surface induced abnormal crystallization” is very interesting, and we are still working on this part. In revised manuscript, the word “COF-366” was thoroughly replaced by “porphyrin-based COF”.

The synthetic procedure of SiO₂@LZU-1 was provided on page S4 of the revised SI: For the synthesis of SiO₂@LZU-1, 6 mg of PEI modified SiO₂ nanospheres and 40.0 mg of PVP were added into 10.0 mL mixed solution of *o*-dichlorobenzene/*n*-butyl alcohol (v/v = 1:1). The dispersion was treated with ultrasonic for more than 0.5 h. Then, 3.5 mg of triformylbenzene and 3.5 mg of *p*-phenylenediamine were added. After stirring for 5 min, 100 μL of glacial acetic acid was added. The reaction was maintained at room temperature for 4 h. Afterward, 400 μL of acetic acid and 480 μL of DI water was added. The mixture was degassed by three freeze-pump-thaw cycles, sealed under vacuum, and kept at 120 °C for 5 days. The products were obtained and washed with acetone by centrifugation.

Also, we have added the following descriptions on Page 13 of the revised manuscript: It is worth mentioning that the XRD pattern of the SiO₂@porphyrin-COF is quite different from the COF-366 (even under the same synthesis conditions). It is speculated that crystalline of porphyrin COF on the spherical nanoparticles with a certain curvature could induce the weakened π - π stacking between successive layers compared with bulk layered COFs, leading to their mismatch stacking. This mismatch stacking may further result in the changing of the cell parameters. This “curved surface induced abnormal crystallization” is very interesting, and we are still working on this part.

Correspondingly, we have revised Figure 3 and added Figure S18 in the revised manuscript and SI, respectively:

Figure 3. Structural unit of (a) LZU-1 and (b) porphyrin-based COF. TEM images of the core@shell structured (c) SiO₂@LZU-1 and (d) SiO₂@porphyrin-COF nanocomposites. Scale bars are 100 nm in c and 200 nm in d.

Figure S18. (A) SEM image and XRD pattern of prepared core@shell structured (c) SiO_2 @LZU-1, scale bar is 400 nm. The obtained nanoparticles are uniform and monodisperse. The XRD pattern shows the typical diffraction peak at 4.7° , indicating the crystallization of LZU-1 COF shell.

2. More characterizations such as FT-IR and solid-state ^{13}C NMR should be applied for providing more information on chemical structures of COFs, especially the imine linkage.

Response: Thanks for the useful suggestion. We have accepted it. The FTIR spectra and solid-state ^{13}C NMR of prepared samples were presented in the revised manuscript (**Figure S10 and S11**). For SiO_2 @TAPB-DMTP-COF, the typical $\text{C}=\text{N}$ stretching vibration at 1618 cm^{-1} confirms the formation of imine linkage. Moreover, the peak at 1677 cm^{-1} can be assigned to the free $-\text{CHO}$ groups due to the existence of bonding effects in the TAPB-DMTP-COF (*ACS Nano* 2019, 13, 11, 13304–13316). The peaks at 2955 and 2842 cm^{-1} are associated with the $\text{C}-\text{H}$ vibration of $-\text{CH}_3$ groups. The ^{13}C NMR of TAPB-DMTP-COF coated SiO_2 also indicates the characteristic peaks at 104, 111, 117, 123, 137, 150, and 203 ppm. In particular, the peaks at 150 ppm can be attributed to the carbon from the Schiff base bonds (*Cell Rep. Phys. Sci.* 2020, 1, 100062). The peak at 171 ppm can be attributed to the carbonyl signal of PVP (*Angew. Chem. Int. Ed.* 2019, 58, 8670–8675).

Correspondingly, we have added the following description on page 8 of the revised manuscript: The small angle X-ray scattering (SAXS) also indicates the porous structure of prepared SiO_2 @COF after thermal crystallization (**Figure S9**). In addition, the imine-linked COF was also characterized by Fourier transform infrared (FTIR) spectra and ^{13}C NMR spectrum. After coated COF onto SiO_2 , the typical $\text{C}=\text{N}$ stretching vibration at 1618 cm^{-1}

appears in the FTIR spectrum of SiO₂@COF, suggesting the formation of imine linkages in the framework of the COF shell (**Figure S10**). As shown in the solid-state ¹³C NMR, the peaks at 104, 111, 117, 123, 137, 150, and 203 ppm are assigned to the aromatic carbon signals from the COF shell (**Figure S11**). In particular, the peaks at around 150 ppm can be attributed to the carbon from the Schiff base bonds.³³ The peak at 171 ppm can be attributed to the carbonyl signal of PVP.³⁴

We have also added Figure S10 and Figure S11 on page S13 of the revised SI:

Figure S10. FTIR spectra of prepared samples. In comparison to the bare SiO₂ and PEI modified SiO₂, the synthesized SiO₂@COF displays extra peaks at 1677 and 1618 cm⁻¹, which can be assigned to the -CHO groups and C=N stretching vibration in the COF shell, respectively. The peaks at 2955 and 2842 cm⁻¹ are associated with the C-H vibration of -CH₃ groups in DMTP.

Figure S11. Solid-state ¹³C NMR spectrum of prepared SiO₂@COF. The characteristic peaks at 104, 111, 117, 123, 137, 150, and 203 ppm is assigned to the aromatic carbon of COF units, suggesting the formation of imine-based COF on SiO₂. The peak at 171 ppm can be attributed to the carbonyl signal of PVP.

3. In the tumor therapy part, comparative experiments using porphyrin-based COF without UCNPs should be performed to tell the necessity of the UCNPs core and the importance of the method.

Response: Thanks for the comment. We have accepted it. Benefiting from the upconverting

property of UCNPs, the combination of UCNPs with porphyrin-based COF enables NIR light-activated PDT. Numerous works have reported that NIR light exhibits deeper tissue penetration than visible light for PDT (*J. Am. Chem. Soc.* 2020, 142, 6822–6832. *Nat. Commun.* 2019, 10, 4416. *Adv. Mater.* 2018, 30, 1802808. *Angew. Chem. Int. Ed.* 2015, 127, 8223–8227). To demonstrate this, we compared the tissue penetration depth of NIR-induced PDT versus visible light triggered PDT side by side. The cancer cells were treated with UC-COF and exposed to 980 and 660 nm laser with or without the intervention of simulated tissue. When the samples were directly irradiated by the laser, the photodynamic effect triggered by the 660-nm laser can kill more cancer cells than the NIR light-activated PDT. However, the killing efficiency decreases by ~75% for the 660 nm-triggered PDT when the light source was blocked by a 4 mm-thickness simulated tissue. In sharp contrast, for the NIR laser-triggered PDT, the killing efficiency only reduces by ~14%. The same tendency was also observed in the live-dead staining images. Even though the cancer cell killing efficiency of the UC-COF group under NIR irradiation is lower than that of the mere COF group under visible light irradiation, the former one offers more favorable tumor treatment in deep tissue.

We have added the following description on Page 14 of the revised manuscript: Compared with the traditional visible light triggered PDT, this UC-COF based NIR-triggered PDT shows much deeper tissue penetration, which is more favorable for tumor treatment in deep tissue (**Figure S22**).

We have also added Figure S22 on Page S21 of the revised SI:

Figure S22. (a) Schematic illustration of the designed experiment for PDT with or without the

intervention of simulated tissue (1 % Intralipid). (b) Cell viability of 4T1 cells after treated with UC-COF under the irradiation of 980 or 660 nm laser, with or without the block of a 4 mm-thickness simulated tissue. Data are presented as mean \pm SD (n = 3). (c) The corresponding live-dead staining images of the treated cells in different groups, scale bars are 200 μ m. When the samples were directly irradiated by the laser, the photodynamic effect triggered by the 660-nm laser can kill more cancer cells than the NIR light-activated PDT. However, the killing efficiency decreases by \sim 75% for the 660 nm-triggered PDT when the light source was blocked by a 4 mm-thickness simulated tissue. In sharp contrast, for the NIR laser-triggered PDT, the killing efficiency only reduces by \sim 14%. The same tendency was also observed in the live-dead staining images. Even though the cancer cell killing efficiency of the UC-COF group under NIR irradiation is lower than that of the mere COF group under visible light irradiation, the former one offers more favorable tumor treatment in deep tissue.

4. Some notes (e.g. Figure S5, S6, S8, and 3c) are not corresponding to the right figures, which should be double-checked and revised.

Response: Thanks very much for the comments. The notes were corrected in the revised manuscript.

5. There are some grammar mistakes. Page 5, line 4, “be overcame” should be revised as “be overcome”; page 7, line 8, “reveal” should be revised as “reveals”.

Response: We appreciate the reviewer’s suggestions. The mistakes pointed out by the reviewer have been revised. We have also checked through the manuscript and made several corrections.

Reviewer #2:

This manuscript proposed a versatile dual-ligands assistant encapsulation strategy for the synthesis of COF-based nanocomposites with customizable structures and functional integration, which provides advancement in the preparation of inorganic nanomaterials functionalized COF nanocomposites. By precisely manipulating the nucleation and growth kinetics of COF, diverse structures of COF-based nanocomposites, including hollow, bowl-shape, yolk-shell, and core-satellites-shell, were successfully fabricated. The COFs were endowed with abundant optical, electrical and magnetic properties. By developing the UC-COF with unique NIR to visible upconverting luminescence, the predicament faced by tradition porphyrin-COFs in the visible light activated PDT with insufficient tissue penetration depth was overcome. Overall, this work has been well-organized, and the results are very interesting, which may attract broad interests from multidisciplinary research communities for different applications. Therefore, it could be considered for publication after addressing the following issues.

Response: We thank the reviewer very much for the positive comments.

1. The dual-ligands clearly played pivotal role in regulating the critical nucleation concentration of COF shell on the surface of SiO₂. Did the dual-ligands also affect the growth rate? What is the exact mechanism of controlling shell thickness by the ligand concentration as shown in Figure 6c?

Response: Thanks very much for the comment. The dual-ligands not only can affect the nucleation of COF on SiO₂, but also affect the growth rate of the COF shell. As we described in the manuscript, both the kinetics of surface nucleation and growth is strongly associated with the ratio of the dual-ligands. When PEI is modified onto SiO₂ surface, the aldehyde monomers can be readily enriched onto the particle surface according to Schiff base reaction. So, the growth rate of the COF shell is accelerated by the PEI ligand dramatically, which inevitably induce the agglomeration of particles during the coating process. When PVP are introduced, the over-speed growth on the PEI modified surface can be greatly suppressed. The decelerating of the growth speed of the COF shell can be attributed to strong interaction between the PVP and the protonated imine units (under acidic condition) in the COF framework, which further result in the passivation of the COF shell (*J. Am. Chem. Soc.* 2017, 139, 13166–13172). So, the growth speed of the COF shell can be well manipulated by tuning the ratio between the PEI and PVP. The shell thickness also can be adjusted from 16 nm to 45 nm by tuning the ratio between the dual-ligands (**Figure 6c and S34**).

We have added the following description on Page 21 of the revised manuscript: When PVP are co-modified on the surface, the over-speed growth on the PEI single ligand modified surface can be greatly suppressed. The decelerating of the growth speed of the COF shell can be attributed to strong interaction between the PVP and the protonated imine units (under acidic condition) in the COF framework, which further result in the passivation of the COF shell.⁴⁶ The growth speed of the COF shell can be well manipulated by tuning the ratio between the PEI and PVP. The shell thickness also can be adjusted from 16 nm to 45 nm by tuning the ratio between the dual-ligands (**Figure 6c and S34**).

2. The critical nucleation concentration was regulated by tuning the interaction between monomers and SiO₂ surface. Whether the concentration of single ligand (PEI or PVP alone) could also regulate the critical nucleation concentration?

Response: We appreciate the reviewer's comments. Actually, PEI is modified onto SiO₂ surface, so the concentration of PEI is supposed to be unchanged during the reaction. To investigate the effect of single ligand of PVP on the growth of COF shell, the concentration of PVP in the solution was changed in the reaction system. TEM images show that although the COF shells can be coated onto SiO₂ nanospheres, but this coating is the non-uniform coating. It can be seen that the COF domains on the SiO₂ nanospheres are "islands", which result in the irregular morphology of COF shell and even partially naked surface of the core. On the other hand, it is very difficult to tune the shell thickness by adjusting the PVP concentration.

We have added the description on Page 20 of the revised manuscript: In the case of PVP modified SiO₂, the thickness of the COF shell is obviously thinner than that of on the PEI-modified SiO₂ (**Figure S33**). Although the COF shells can be coated onto SiO₂ nanospheres, but this coating is the non-uniform coating. It can be seen that the COF domains on the SiO₂

nanospheres are “islands”, which result in the irregular morphology of COF shell and even partially naked surface of the core. On the other hand, it is very difficult to tune the shell thickness by adjusting the PVP concentration.

We have added Figure S33 on Page S27 of the revised SI:

Figure S33. TEM images of the obtained SiO₂@COF in the PVP solution at different concentration: a) 2 mg/mL, b) 4 mg/mL, c) 10 mg/mL, d) 20 mg/mL and e) 40 mg/mL. Scale bar are 200 nm. With the absence of PEI, the COF domains on the SiO₂ nanospheres are “islands”, which result in the irregular morphology of COF shell and even partially naked surface of the core. On the other hand, it is very difficult to tune the shell thickness by adjusting the PVP concentration.

3. Could the modification sequence of the dual ligands affect the final products? Please provide detailed description in the text.

Response: Thanks for the useful comments. As we described in the manuscript, the as-prepared cores were first modified with PEI and dispersed in the PVP solution, during which the PVP can attach onto the surface of cores. Thereafter, the polymerization and thermal crystallization of imine monomers were proceeded. The PVP in reaction solution is essential for regulating the growth kinetics of COF shell on the PEI modified surface. If the modification sequence and manner of PEI and PVP are reversed, the COF shell is unable to deposit on PVP-modified SiO₂ in the PEI solution. Thus, the modification sequence of the dual ligand cannot be changed.

Figure. TEM image of samples prepared by coating COF shell onto PVP-modified SiO₂ in the PEI-contained reaction system (2 mg/mL). Scale bar is 100 nm. If the modification sequence

and manner of PEI and PVP are reversed, the COF shell is unable to deposit on PVP-modified SiO₂ in the PEI solution, and only bare SiO₂ nanospheres were obtained.

4. The bowl and yolk-shell structure were obtained by selecting etching of SiO₂ segment. The author should investigate whether the porous structure of COF shell was changed or not after the alkaline etching process.

Response: We appreciate the reviewer for the constructive comment. We have accepted it. The porous structure of bowl and yolk-shell structured COF nanoparticles were investigated by TEM, XRD and N₂ adsorption-desorption. The N₂ adsorption-desorption isotherms of bowl and yolk-shell structured COF nanoparticles were presented in the revised manuscript. The specific surface area of the resultant bowl and yolk-shell structured COF nanoparticles shows no obvious change after the etching process. Moreover, the XRD patterns of bowl and yolk-shell structured COF nanoparticles were also measured and presented. All diffraction peaks of TAPB-DMTP COF are reserved. The typical diffraction signals corresponding to UCNPs are also detected. The results indicate that the porous structure of bowl and yolk-shell structure is not changed after the etching process.

We have added Figure S14 on Page S15 of the revised SI:

Figure S14. (A) N₂ adsorption-desorption isotherms and pore distribution (inset image), and (B) XRD patterns of the synthesized bowl-shaped COF nanoparticles and yolk-shell structured UCNPs@COF in Figure 2d and 2f, respectively. The asterisk labeled peaks are assigned to the typical diffraction peaks of UCNPs. The results show that the porous structures of COF shell are not changed after the selective alkaline etching treatment, implying the high stability of prepared COF-based nanocomposites.

5. The UCNPs were also pre-coated with SiO₂ for the synthesis of UC-COF, but SiO₂ seemed to be useless for the biomedical application of UC-COF. The authors should give a brief explanation about this.

Response: Thank you very much for the comments. Actually, the as-synthesized upconversion nanocrystal NaYF₄:Yb/Er@NaYF₄ is capped with hydrophobic oleic acid ligands. The coating

of SiO₂ is a convenient way to transfer the hydrophobic UCNPs to water, facilitating the further modification of dual-ligands for the growth of COF shell.

We have added the description on Page 13 of the revised manuscript: Because of the hydrophobic surface of the obtained UCNPs, a dense SiO₂ layer was coated on the UCNP to form the hydrophilic core@shell UCNP@SiO₂ nanoparticles, which greatly facilitates the further modification of dual-ligands and the controlled growth of COF shell.

6. The colloidal stability of UC-COF in different media should be evaluated by DLS.

Response: We appreciate this suggestion. We have accepted it. The size distributions of UC-COF in water, saline and fetal bovine serum (FBS) were measured by DLS. The results demonstrate that the average diameter of UC-COF maintains well in different media, indicating the good colloidal stability of UC-COF. The photographs of UC-COF dispersions in different media did not show any agglomeration or precipitations after 12 h incubation.

We have added the following illustration on Page 14 of the revised manuscript: The UC-COF can be stabilized with PVP due to the interaction between PVP and imine units in the COF framework, endowing the synthesized UC-COF with good colloidal stability for biomedical applications (**Figure S20**).

We have also added Figure S20 on Page S20 of the revised SI:

Figure S20. Size distribution of UC-COF dispersed in different media for (a) 1 h and (b) 12 h. Inset represents the digital photos of the corresponding dispersions. No obvious agglomeration or precipitations was observed after 12 h incubation in different physiological conditions, suggesting the good colloidal stability of prepared UC-COF.

7. The authors need to check the full text carefully, e.g., the caption for Figure 2f is missing, in the caption of Figure 5, “Scale bars in d” should be “Scale bars in e”.

Response: Thanks for the comments. We have accepted it and the whole text has been carefully revised.

We have added the caption of Figure 2f on Page 10 of the revised manuscript: (f) TEM image of core@satellite@shell structured Zr-MOF@DCNPs@COF.

Reviewer #3:

This is another excellent work from the mesopore group. The authors target the key challenges in the composition of COFs with inorganic functional materials, i.e. the

mismatch at the interface of these two different classes of materials. Different from previous works that using surface modification of small organic molecules, the method used here involves the combinational use of two types of polymers, PEI rich in amino functional groups, and PVP rich in aldehyde. These two polymers offer strong chemical interactions with two building blocks of the COF separately, therefore generate a robust interface for the adhesion and growth of COF coating layer. The advantage of using two kinds of polymers, instead of small molecules with the same functional group, is that the conformation of the polymers avoids intermolecular reaction between the amino and aldehyde functional groups, thus providing sufficient anchors for the COF building blocks at the interface. This combined with the relatively well established polymer coating on inorganic functional materials, such as metal oxides, makes this new method applicable for the composition of COF with a large variety of inorganic cores. In general, this is an important advance in the synthesis of state-of-the-art COF based functional materials, and it is highly recommended for publication.

Response: We appreciate the reviewer very much for the positive comments.

1. Is there any way to quantify the thickness of the polymer layers applied prior to the growth of COFs? This is critical for the reproduction of results and the quality control.

Response: Thank you very much for the suggestion. We have accepted it. Actually, the thickness of the polymer layers on SiO₂ is almost invisible in TEM (**Figure S1**). The thermogravimetric (TG) analysis was used to quantify the content of polymer on SiO₂. Three samples, including bare SiO₂, PEI modified SiO₂, and the PEI/PVP co-stabilized SiO₂ were selected for analysis. The results show that the weight loss of the samples was gradually increased in order from SiO₂ (10.6%), SiO₂-PEI (12.2%) to SiO₂-PEI/PVP (13.5%), suggesting the successive attachment of the ligands. So, the amount of the ligands on the SiO₂ is about 1.6 % and 1.3 % for PEI and PVP, respectively.

We have also added Figure S31 on Page S26 of the revised SI:

Figure S31. TG curves of bare SiO₂, PEI modified SiO₂ and dual ligands modified SiO₂ nanospheres. The result shows that the weight loss of the samples was gradually increased in order from SiO₂ (10.6%), SiO₂-PEI (12.2%) to SiO₂-PEI/PVP (13.5%), suggesting the

successive attachment of the dual ligands onto SiO₂ surface. The amount of the ligands on the SiO₂ is about 1.6 % and 1.3 % for PEI and PVP, respectively.

2. The orientation of the COF crystals should be further discussed. TEM figures only provide the projected view of each nanoparticle. The fringes in the high resolution images assigned to the COF crystal, could either reflect the interlayer distance between the layers of this 2D COF, or the size of the vertical pores perpendicular to the 2D plane of the COF layers, depending on the orientation of the COFs. Both of these two possibilities should be assessed and further studied. The orientation of the pores, perpendicular, or parallel to the spherical particle surface will influence its uptake behavior for the guest molecules.

Response: We appreciate the reviewer very much for the insightful comments. In the revised manuscript, the HRTEM were introduced to investigate the orientation of the COFs. The fringes with an interplanar spacing of ~ 2.8 nm in the HRTEM image belong to the vertical pores perpendicular to the 2D plane of the COF layers. Previous reports have reported that TPB-DMTP-COF adopts the AA stacking mode with an interlayer distance (*c*) of 0.35 nm (*Chem. Commun.* 2016, 52, 3690-3693. *Nature Chem.* 2015, 7, 905–912). As evidenced by the appearance of the hexagonal pores perpendicular to the SiO₂ surface in HRTEM image, it can be speculated that the 2D plane of COF layers parallel to the SiO₂ surface. (**Figure S4**).

We have added the following description on Page 7 of the revised manuscript: Thus, we can conclude that the 2D plane of the COF layers is parallel to the SiO₂ surface, and the fringes with an interplanar spacing of ~ 2.8 nm can be assigned to the periodic pores perpendicular to the 2D plane of the COF layers and the SiO₂ surface (**Figure S4**).^{3,29}

We have added the Figure S4 on Page S10 of the revised SI:

Figure S4. (a, b) HRTEM image of single SiO₂@COF nanoparticle. Scale bars are 100 nm in (a) and 5 nm in (b). The fringes belong to the vertical pores perpendicular to the 2D plane of the COF layers. The projected view of the ~ 2.8 nm pores can also be clearly observed. So, we can conclude that the 2D plane of the COF layers is parallel to the SiO₂ surface, and the fringes can be assigned to the periodic pores perpendicular to the 2D plane of the COF layers and the SiO₂ surface.

3. It is known that the domain size of COF crystals and band structure are important for the optical behavior (*Angew. Chem. Int. Ed.*, 2019, 58, 14213, *Matter*, 2020, 2, 4, 1049, *Aggregate*, 2021, <https://doi.org/10.1002/agt2.24>). Investigation on these molecular aspects of the COF coating will help to further understand the optical properties of the COF composite.

Response: We appreciate the reviewer's comment and have accepted it. In the revised manuscript, the band gap of the core@shell structured COF nanocomposites was calculated based on Kubelka-Munk theory. Similar with the literature reports, the core@shell structured COF nanocomposites exhibited smaller band gaps compared with the corresponding molecular building blocks (**Figure S12**), demonstrating the cooperation between chromophores across the entire COF crystals. We did not observe any other abnormal optical behavior for the core@shell structured COF nanocomposites (e.g. PDT, two-photon absorption, fluorescence). The related results have been provided in the revised manuscript.

We have added the following description on Page 8 in the revised manuscript: Since the optical property of COF-based materials has attracted increasing attention recently, the band gap of SiO₂@COF was estimated based on the UV-vis reflectance spectra to investigate the optical property of SiO₂@COF (**Figure S12**). Similar with the literature reports,^{35, 36, 37} the SiO₂@COF nanocomposites exhibited smaller band gaps compared with the corresponding molecular building blocks, demonstrating the cooperation between chromophores across the entire COF crystals.

We have added the Figure S12 on Page S14 of the revised SI:

Figure S12. (a) UV-vis reflectance spectra and (b) the band gap of different samples estimated from the UV-vis reflectance spectra. The results show that the SiO₂@TAPB-DMTP COF nanocomposites exhibit smaller band gaps compared with the corresponding molecular building blocks, demonstrating the cooperation between chromophores across the entire COF crystals.

4. Another interesting aspect is the capability to produce Yolk shell shaped COF composite. It might worth of discussing the advantage of such structures with large inside cavity and small pore apertures in the shell.

Response: Thank you very much for the suggestion. We have accepted it. The prospects of the yolk-shell structured COF nanocomposites were discussed in the revised manuscript.

We have added the following description on Page 9 of the revised manuscript: The yolk-shell structured COF nanocomposites are expected to exhibit many advantages, such as the high loading capacity, controllable releasing kinetics for cargos, unique spatial confinement effect for catalysis *etc.*³⁹

REVIEWERS' COMMENTS

Reviewer #1 (Remarks to the Author):

The questions have been well addressed, and I agree to publish this manuscript as it is.

Reviewer #2 (Remarks to the Author):

The authors have revised the manuscript as the referee suggested, therefore, it could be considered for publication in its current form.

Reviewer #3 (Remarks to the Author):

The authors have made efforts to appropriately address all the scientific concerns. It is a top notch work and is highly recommended for publication.